# 16S-23S rRNA Internal Transcribed Spacer Region (*ITS*) Sequencing: A Potential Molecular Diagnostic Tool for Differentiating *Lactococcus garvieae* and *Lactococcus petauri*

**DOI:** 10.3390/microorganisms11051320

**Published:** 2023-05-17

**Authors:** Nadia Stoppani, Silvia Colussi, Paolo Pastorino, Marino Prearo, Simona Sciuto, Ilhan Altinok, Rafet Çağrı Öztürk, Mustafa Ture, Ana Isabel Vela, Maria del Mar Blanco, Charalampos Kotzamanidis, Konstantina Bitchava, Andigoni Malousi, Lucio Fariano, Donatella Volpatti, Pier Luigi Acutis, Jose Francisco Fernández-Garayzábal

**Affiliations:** 1Istituto Zooprofilattico Sperimentale del Piemonte, Liguria e Valle d’Aosta, 10154 Turin, Italy; nadiastop96@gmail.com (N.S.); paolo.pastorino@izsto.it (P.P.); marino.prearo@izsto.it (M.P.); simona.sciuto@izsto.it (S.S.); pierluigi.acutis@izsto.it (P.L.A.); 2Faculty of Marine Sciences, Karadeniz Technical University, Sürmene, 61530 Trabzon, Turkey; ialtinok@gmail.com (I.A.); rafetcagriozturk@gmail.com (R.Ç.Ö.); 3Central Fisheries Research Institute (SUMAE), 61250 Trabzon, Turkey; mustafa61ture@gmail.com; 4VISAVET and Department of Animal Health, Universidad Complutense de Madrid, 28040 Madrid, Spain; avela@ucm.es (A.I.V.); mmblanco@ucm.es (M.d.M.B.); jffernandez@vet.ucm.es (J.F.F.-G.); 5Veterinary Research Institute, ELGO-DIMITRA, 54124 Thessaloniki, Greece; kotzam@vri.gr; 6School of Animal Biosciences, Agricultural University of Athens, 11855 Athens, Greece; bitchava@aua.gr; 7Laboratory of Biological Chemistry, Medical School, Aristotle University of Thessaloniki, 54124 Thessaloniki, Greece; andigoni@auth.gr; 8Azienda Agricola Canali Cavour, 12044 Centallo, Italy; luciof@libero.it; 9Department of Agricultural, Food, Environmental and Animal Sciences (DI4A), University of Udine, 33100 Udine, Italy; donatella.volpatti@uniud.it

**Keywords:** *Lactococcus garvieae*, *Lactococcus petauri*, genome, 16S-23S internal transcribed spacer region, diagnostic technique, lactococcosis

## Abstract

*Lactococcus garvieae* is the etiological agent of lactococcosis, a clinically and economically significant infectious disease affecting farmed rainbow trout. *L. garvieae* had been considered the only cause of lactococcosis for a long time; however, *L. petauri*, another species of the genus *Lactococcus*, has lately been linked to the same disease. The genomes and biochemical profiles of *L. petauri* and *L. garvieae* have a high degree of similarity. Traditional diagnostic tests currently available cannot distinguish between these two species. The aim of this study was to use the transcribed spacer (ITS) region between 16S rRNA and 23S rRNA as a potential useful molecular target to differentiate *L. garvieae* from *L. petauri*, saving time and money compared to genomics methods currently used as diagnostic tools for accurate discrimination between these two species. The *ITS* region of 82 strains was amplified and sequenced. The amplified fragments varied in size from 500 to 550 bp. Based on the sequence, seven SNPs were identified that separate *L. garvieae* from *L. petauri*. The 16S-23S rRNA ITS region has enough resolution to distinguish between closely related *L. garvieae* and *L. petauri* and it can be used as a diagnostic marker to quickly identify the pathogens in a lactococcosis outbreak.

## 1. Introduction

Lactococcosis is a serious septicemic fish disease in Mediterranean countries. *Lactococcus garvieae* was considered the sole causative agent of lactococcosis until recently. *L. garvieae* has been isolated as a causative agent of the disease in several freshwater and marine species such as rainbow trout (*Oncorhynchus mykiss*), tilapia (*Oreochromis* sp.), Japanese eels (*Anguilla japonica*), olive flounders (*Paralichthys olivaceous*), grey mullet (*Mugil cephalus*), catfish (*Ameiurus* spp.), wild wrasses (*Coris aygula*), black rockfish (*Sebastes schlegeli*), amberjacks (*Seriola dumerili*), kingfish (*Seriola lalandi*), and giant freshwater prawns (*Macrobrachium rosenbergii*). Mortalities associated with the disease have been reported in America, Africa, Europe, Asia, and Oceania [1].

It is one of the most important diseases in farmed rainbow trout, causing serious economic losses with high mortality rates [2]. Other species such as the common carp (*Cyprinus carpio*) seem to be resistant to the disease.

There are several aquatic environmental factors such as fish stress, overcrowding, mishandling, poor water quality, and water temperature that influence the occurrence of lactococcosis outbreaks. Indeed, water temperature is one of the most important predisposing factors that increases susceptibility to infection. Most acute outbreaks occur when the water temperature is over 18 °C, although acute outbreaks have been described at a water temperature of 14–15 °C. Moreover, a decreased oxygen availability in the water can emerge, leading to stressful conditions for trout and increasing their susceptibility to infection [3]. Transmission of the disease occurs by horizontal mechanisms, mainly through water, fish injuries, and by the fecal–oral route [1]. Infected fish exhibit nonspecific clinical signs such as lethargy, erratic swimming, dark skin pigmentation, loss of orientation, early anorexia, and a marked monolateral or bilateral exophthalmia [1]. Macroscopic signs include external and internal hemorrhages (mainly in the eyes, liver, and intestine), splenomegaly, and cerebral congestion. This pathogen causes direct economic losses due to elevated rates of mortality (up to 50%), but also indirect economic losses due to the decrease in growing rates, the increased labor for fish management, and treatment costs. As a result of the development of antibiotic resistance, the use of chemotherapeutic agents in the control of lactococcosis is an unsustainable strategy. In trout farms, preventative biosecurity measures such as the destruction of dead fish, regular and appropriate disinfection of equipment, improvement of health management measures, and immunization of healthy fish are highly recommended [1].

*Lactococcus petauri* was first isolated from a facial abscess in a sugar glider (*Petaurus breviceps*) by Goodman et al. [4], who characterized it as a new member of the genus *Lactococcus*. The first report of an *L. petauri* outbreak in fish was described in farmed rainbow trout imported from Spain to Greece [5]. The pathogen was initially identified as *L. garvieae* before being reclassified as *L. petauri* in a retrospective analysis. Fish infected by *L. petauri* exhibit the same clinical signs and symptoms as *L. garvieae*-infected fish. As a result, both species can be considered causative agents of lactococcosis in fish. Recent retrospective studies have revealed that *L. petauri*, not *L. garvieae,* is responsible for the majority of lactococcosis outbreaks. In recent years, an outbreak of lactococcosis caused by *L. petauri* in California, USA, resulted in >50% mortality rate and the culling of more than 3.2 million farmed rainbow trout [6]. Moreover, in Brazil, an outbreak of lactococcosis caused by *L. petauri* resulted in a high mortality rate in farmed Nile tilapia (*Oreochromis niloticus*) [7]. The real relevance of *L. petauri* as a bacterial pathogen for Nile tilapia is still to be assessed, considering that in recent years, infections of *L. garvieae* have been frequently reported; however, this was before *L. petauri*’s description [7]. Based on these findings, greater attention should be given to this novel pathogen.

Lactococcosis is traditionally diagnosed in the laboratory by using conventional microbiological techniques or commercial phenotypic and biochemical tests. The most widely used commercial phenotypic method for a routinary identification of bacteria is the API (Analytical Profile Index). The API identifies bacteria based on sugar fermentation (carbohydrates), the assimilation of certain other carbon sources, and the production of certain unique metabolites and enzymes. Both *L. garvieae* and *L. petauri* are lactic acid bacteria (LAB) and they have similar phenotypic and biochemical properties [4]. Matrix-assisted laser desorption ionization time-of-flight mass spectrometry (MALDI-TOF MS) generally allows an accurate identification at the species level of most microorganisms and can be an important technique to rapidly identify etiological agents [8]. MALDI-TOF MS, however, does not distinguish between these two species [7].

Advantages in molecular methods have provided a novel set of tools that characterize pathogens in ever greater detail [9]. The 16S rRNA gene has been widely used in medical microbiology to identify bacteria [10]. The sequence contains hypervariable regions that can provide species-specific signature sequences useful for bacteria identification due to the presence of divergent regions in the 16S rRNA gene among various bacteria, while conserved regions allow bacteria to be categorized within the same species [11]. A species-specific PCR assay for the identification of *L. garvieae* was developed by Zlotkin et al. [12]. Yet, a specific 16S rRNA gene analysis performed on *L. garvieae* and *L. petauri* also failed to distinguish between *Lactococcus* species [7,13]. This confirms the literature findings, since the analysis of the 16S rRNA gene is insufficient to distinguish between *L. petauri* and *L. garvieae* [13,14].

Whole genome sequencing (WGS) performed by Kotzamanidis et al. [15] was the first diagnostic technique that allowed differentiation between these two bacterial species. WGS is the most modern technique used for species characterization and identification [16]. It identifies sequence fragments specific to the taxonomic level and provides access to the full genetic content of microorganisms. The average nucleotide identity (ANI) and digital DNA–DNA hybridization (dDDH) values are extensively used for reliable species differentiation [17]. The ANI score exceeded the recommended cut-off value of ~96% and showed the highest similarity (98.393%) with the LG_SAV_20 strain, isolated in 2007 from rainbow trout affected by lactococcosis in Greece [15] with *L. petauri* 159,469, described as a representative of the species [4]. In addition, dDDH revealed an 82.3% (cut-off point of 70%) similarity to *L. petauri* 159,469 [18]. This result led to the classification of the LG_SAV_20 strain as the representative sequence of the new species *L. petauri* isolated from rainbow trout. The adoption of WGS as a routine technique to discriminate between *L. garvieae* and *L. petauri* incurs a high expense per sample. It is therefore unrealistic to use this technique as a routine standard procedure for diagnosing lactococcosis. Developing a new rapid method would pave the way for a faster and more cost-effective diagnosis, making it simpler to use in routine diagnostic practice.

An efficient characterization system applied in Californian and Brazilian outbreaks of *L. petauri* is multilocus sequence typing (MLST); only three genes were used by Shahin et al. [6], these were Carbamate kinase (*arcC*), Gyrase B (*gyrB*), and RNA polymerase B (*rpoB*). Seven genes were instead used for Brazilian outbreaks [7]: α-acetolactate synthase (*als*), α-subunit of ATP synthase (*atpA*), elongation factor EF-Tu (*tuf*), glyceraldehyde-3-phosphate dehydrogenase (*gapC*), DNA gyrase β-subunit (*gyrB*), RNA polymerase β’-subunit (*rpoC*), and galactose permease (*galP*). In particular, the main differences were related to the *gyrB* gene. It presented similarities higher than 99.6% with *L. petauri* references strains (B1726, PAQ102015-99, CF11, and 159469) and 99.9% similarities with *L. garvieae* reference strains (ATCC49156, Lg2, and JJJN1) [7].

The amplification of a short region (290 bp) of *ITS* has already been suggested as a useful tool for *L. garvieae* diagnoses [19]. The *ITS* region has been considered a good tool for species specific bacterial identification among related organisms due to its genetic variability in size and sequence compared to 16S rRNA and 23S rRNA genes [20]. This method was recently used for molecular diagnosis in piscine lactococcosis [6]; however, its role in distinguishing between *L. garvieae* and *L. petauri* has not been reported.

In this study, we used the internal transcribed spacer (*ITS*) region 16S-23S of rRNA as a tool for distinguishing and identifying *L. garvieae* and *L. petauri* isolated from Mediterranean countries such as Greece, Italy, Spain, and Turkey.

## 2. Materials and Methods

### 2.1. Strains

A total of 104 *L. garvieae* and *L. petauri* strains were included in the study (Table 1 and Table 2). Seven strains of *L. garvieae* (*n* = 3) and *L. petauri* (*n* = 4), whose genomes are publicly available, were used as control strains for analyses and further comparison of the 16S-23S rRNA ITS sequences (Table 2). The genomes of the strain types of *L. garvieae* and *L. petauri* were also included in the study.

We also retrieved additional 16S-23S transcribed spacer (*ITS*) region sequences from both *L. garvieae* and *L. petauri* (*n* = 8 and *n* = 5, respectively; reference strains). Sequences of *L. garvieae* were directly retrieved from the NCBI database: *L. garvieae* LMG 9472 (accession numbers HM241914), *L. garvieae* LMG 8162 (accession number HM241916), *L. garvieae* C1 (accession number MZ146920), *L. garvieae* 108-33 (accession number MZ146924), *L. garvieae* 106-30 (accession number MZ146925), *L. garvieae* 168 (accession number MZ146926), *L. garvieae* 20,684 (accession number AF225967), and *L. garvieae* L1-5 (accession number AF225968). Sequences of *L. petauri* strains were retrieved from published whole genome sequences: *L. petauri* LG4 (accession number CP086401; region: 1911306-1911836), *L. petauri* LG26 (accession number CP086595; region 2048225-2048755), *L. petauri* B1726 (accession number CP094882; region 1717542-1718072), *L. petauri* NHH01_13 (accession number JANHCX010000013; region 2696-3716), *L. petauri* LG_SAV_20 (accession number SIVY01000041; region 2699-3719), *L. petauri* LG6 (accession number JAOYNZ010000013; region 2208-3718), *L. petauri* LG3 (accession number JAOYNX010000011; region 1571-2101), *L. petauri* LG5 (accession number JAOYNY010000011; region 1571-2101), and *L. petauri* LG1 (accession number JAOYNW010000014; region 2208-3718).

Afterwards, 82 field clinical strains from Italy (*n* = 22), Turkey, Spain, and Greece (20 strains/each) isolated from rainbow trout (*Oncorhynchus mykiss*) were included in this study (Table 1). Eighty-one isolates were initially identified as *L. garvieae* through both conventional microbiological methods and PCR according to the protocol described by Zlotkin et al. [12]. The strain 20-GR was identified as *L. petauri* based on a genomic analysis [15]. All strains were stored at −80 °C in a cryobank until use. Strains were sent to the Aquatic Biology, Aquaculture, and Fish Disease Laboratory of the Istituto Zooprofilattico of Piemonte, Liguria and Valle d’Aosta. Then, the strains were thawed, reactivated in brain heart infusion (BHI) liquid-enriched medium (Microbiol s.n.c., CA, Italy), and subsequently streaked on Columbia Blood Agar (CBA) and incubated at 22 ± 2 °C for 24 h.

### 2.2. Polymerase Chain Reaction (PCR) of ITS 16S-23S Region

The boiling and freeze–thawing methods were used for DNA extraction: briefly, the colonies of interest were resuspended in DNAse-free water and boiled for 10 min at 95 °C, followed by rapid chilling at −20 °C and DNA pelleting by centrifugation for 1 min at 11,200× *g* [21]. The 16S-23S rRNA *ITS* region was amplified by PCR using the primers 16S 5′-GCTGGATCACCTCCTTTCT-3′ and 23S 5′-GGTACTTAGATGTTTCAGTTCC-3′ described by Kabadjova et al. [22]. PCR was carried out in a final volume of 25 μL, containing 2.5 μL of 10X PCR Buffer (-MgCl_2_), 1 μL of 50 mM MgCl_2_, 0.5 μL of dNTPs (VWR), 0.3 μL of each primer (20 μM), 0.2 μL of Platinum *Taq* DNA polymerase (Invitrogen), and 50 ng of the template DNA. The reference strains *Lactococcus garvieae* DSMZ20684 and *Lactococcus petauri* DMSZ104842 were used as amplification positive controls. Amplifications were performed using the following thermal profile: initial denaturation at 94 °C for 2 min, followed by 32 cycles of denaturation at 94 °C for 1 min, annealing at 56 °C for 1 min, extension at 72 °C for 1 min, and final extension at 72 °C for 10 min. Amplicons were run on a 2% agarose gel for quantification. Amplicons were purified with a Qiaquick purification kit (Qiagen) and bi-directionally sequenced using the Brilliant Dye Terminator (v1.1) Cycle Sequencing Kit (NimaGen) on a genetic analyzer (Applied Biosystems 3130, Thermo Fisher). DNA sequence analyses were performed in DNASTAR Lasergene Software.

A single representative *ITS* sequence for a strain, isolated in different countries, was deposited to GenBank (www.ncbi.nlm.nih.gov/genbank (accessed on 2 January 2023)) under the following accession numbers: 1-IT for Italian strains OQ108343; 1-GR for Greek strains OQ108344; 1-TK for Turkish strains OQ108345; and 1-SP for Spanish strains OQ108346.

### 2.3. Phylogenetic Analysis

The 16S-23S rRNA *ITS* region sequences of *L. garvieae* and *L. petauri* reference strains, reported in Table 2, were retrieved from GenBank and used to perform a neighbor-joining analysis [23] using MEGA X [24]. Evolutionary distances were ascertained via the maximum composite likelihood method [25]. A 1000-replicate bootstrap test was performed. *Lactococcus lactis* strain KCTC 3768 (accession number HM241926) was used as an outgroup.

### 2.4. Genome Sequencing

The genomic DNA of a subset of nine clinical strains (Table 3) from Spain (*n* = 1), Italy (*n* = 2), Turkey (*n* = 2), and Greece (*n* = 4) was extracted using a QIAGEN DNeasy Blood and Tissue kit (Qiagen, Hilden, Germany). The concentration of genomic DNA was quantified using a Qubit^®^ dsDNA BR Assay kit and a Qubit 3.0 Fluorometer (Thermo Fisher Scientific, Waltham, MA, USA). Illumina sequencing libraries were generated with a Nextera XT DNA Library Preparation Kit and sequenced on an Illumina MiSeq platform with 2 × 300-bp paired-end reads (Illumina, San Diego, CA, USA). The genomes of the nine isolates were assembled into contigs and scaffolds using the SPAdes algorithm with default parameters [26], and the quality of the assembly was checked by QUAST software [27].

### 2.5. Genetic Identification

Genome sequences of the strain types of *L. garvieae* (DSM 20684^T^) and *L. petauri* (159469^T^) were retrieved from GenBank (accession numbers JXJV01000001 and NZ_MUIZ0000000, respectively). The Genome-to-Genome Distance Calculator tool (GGDC; https://ggdc.dsmz.de/ggdc.php# (accessed on 8 May 2023)) was used to determine the digital DNA–DNA hybridization (dDDH) between the genomes of the seven control strains, nine field clinical strains, and those of the type strains. In addition, the average nucleotide identity (ANI) values were also calculated using the ANI Calculator tool (https://www.ezbiocloud.net/tools/ani (accessed on 8 May 2023)). Both ANI and dDDH are methods routinely used to delineate bacterial species using threshold values of ≥70% for dDDH and 95–96% for ANI [17,28].

## 3. Results

The alignment of the 16S-23S ITS sequences of the seven control strains and that of *L. garvieae* and *L. petauri* strain types revealed a different SNP pattern for *L. garvieae* and *L. petauri*. Thus, six diagnostic sites distinguishing *L. garvieae* from *L. petauri* were found (Table 2 and Figure 1): an adenine (A) insertion at position g.218_219 insA, 5 SNPs at position g.358G>T, g.440T>G, g.442T>A, g.469C>A, g.478G>T, and a deletion (-) at position g.443 delT. These SNP patterns were consistently detected between *L. garvieae* and *L. petauri* reference and clinical field strains (Table 2). A phylogenetic analysis of the strains included in this study revealed two well-differentiated clusters that included all the *L. garvieae* and *L. petaurid* strains, respectively. Both clusters were supported by significant bootstrap values. Italian strains clustered together with the *L. garvieae* control and reference strains, while Greek, Spanish, and Turkish strains clustered with the *L. petauri* control and reference strains (Figure 2).

The data for dDDH and ANI values for the seven control strains and nine clinical field strains are shown in Table 3. The three *L. garvieae* control strains displayed average dDDH and ANI similarity values with *L. garvieae* DSM20684^T^ of between 85.1 and 99.9% and 98.3 and 99.9%, respectively. The average dDDH and ANI values of the four *L. petauri* control strains with *L. petauri* 159469^T^ ranged between 85.4 and 89.5% and 98.3 and 98.9%, respectively. Among the field clinical strains, both Italian isolates exhibited dDDH and ANI similarity values with *L. garvieae* DSM20684^T^ higher than the threshold values stablished for species delineation. On the other hand, the Spanish, Turkish, and Greek strains exhibited dDDH and ANI similarity values with *L. petauri* 159469^T^ higher than those stablished for species delineation (Table 3).

## 4. Discussion

*L. garvieae* had been considered until recently the only etiological agent of lactococcosis [1]. However, recent studies have demonstrated the implication of a new lactococcal species, *L. petauri*, in the etiology of this disease [4,5,6,7]. Therefore, both *L. petauri* and *L. garvieae* are responsible for lactococcosis outbreaks. Differentiation of both species in common microbiological diagnostic laboratories is difficult due to their phenotypic and genetic similarity, which results in identification errors. Thus, commercial identification systems such as MALDI-TOF or PCR assays targeting the 16S rRNA gene cannot discriminate between isolates of both species [7,13,14]. Nevertheless, the interspecies divergence in the 16S–23S ITS sequences makes this region a potentially good candidate for bacterial discrimination and identification. Therefore, in this study, we investigate the polymorphism in the 16S rRNA and 23S rRNA transcribed spacer (ITS) region as a potentially useful molecular target to differentiate *L. garvieae* from *L. petauri*.

As there was the possibility that the identification of some of the *L. garvieae* and *L. petauri* strains available in public databases was not correct, the identification of a subset of seven strains (control strains) was corroborated by calculating their dDDH and ANI values by comparing their genomes with those of the *L. garvieae*- and *L. petauri*-type strains. The control strains displayed dDDH and ANI values higher than the threshold values considered for species delineation with their respective species, confirming therefore their correct identification (Table 2 and Table 3). Their 16S-23S ITS sequences were aligned, revealing two different ITS patterns for *L. garvieae* and *L. petauri* characterized by differences in five SNP and two indel variations (Table 2). Later on, these variations were confirmed in all the reference and field clinical strains when their 16S-23S ITS sequences were matched to those of the control strains (Table 2). Thus, the reference strains exhibited an ITS pattern according to their available identification. Regarding clinical field strains, the Italian isolates exhibited the ITS pattern characteristic of *L. garvieae*, while the Spanish, Greek, and Turkish isolates displayed the ITS pattern of *L. petauri*. A phylogenetic analysis based on the 16S-23S ITS sequences revealed two well-defined clusters (Figure 2), one cluster including all control, reference, and field clinical strains of *L. garvieae*, while the other cluster included all *L. petauri* strains. These data indicate that the identification available in GenBank of the reference strains is correct. Regarding clinical field strains, the Italian strains correspond to *L. garvieae* while the Spanish, Turkish, and Greek strains correspond to *L. petauri*. These results are consistent with previous results based on WGS and confirm the relevant role of *L. petauri* in the epidemiology of lactococcosis in some Mediterranean countries [5,13]. Moreover, genetic identification results support the identification based on the analysis of the 16S-23 ITS sequences. Thus, the genomes of the seven control and nine clinical field strains analyzed in this study showed respective strain type dDDH and ANI values higher than the threshold values stablished for species delineation (≥70% for dDDH and 95–96% for ANI; Table 3), confirming therefore their identification. This genetic identification was performed in about 11% of the *L. garvieae* and *L. petauri* clinical strains, but the clear phylogenetic differentiation of *L. garvieae* and *L. petauri* based on their 16S-23 ITS sequences together with the consistence of their different SNP patterns makes it reasonable to assume the identification for the remaining field clinical strains.

An analysis based on the sequencing of the *gyrB* gene, already used as a phylogenetic marker for several closely related genera [29], has been proposed for differentiating *L. petauri* and *L. garvieae*, grouping isolates of these species in different clusters [6,7]. In this sense, sequencing of the 16S-23S ITS regions should be as accurate, efficient, and cost effective as *gyrB* sequencing. Overall, the results of this study indicate the great potential of the identified SNP markers for the differentiation of *L. garvieae* from *L. petauri.*

## 5. Conclusions

The results of this study show that *L. garvieae* and *L. petauri* strains display two well-differentiated patterns in their 16S-23S ITS regions, characterized by five SNP differences and two indel variations that were consistently identified in all the strains investigated. Therefore, sequencing of the 16S-23S rRNA ITS region can be a useful tool for discriminating between *L. garvieae* and *L. petauri*.

## Figures and Tables

**Figure 1 microorganisms-11-01320-f001:**
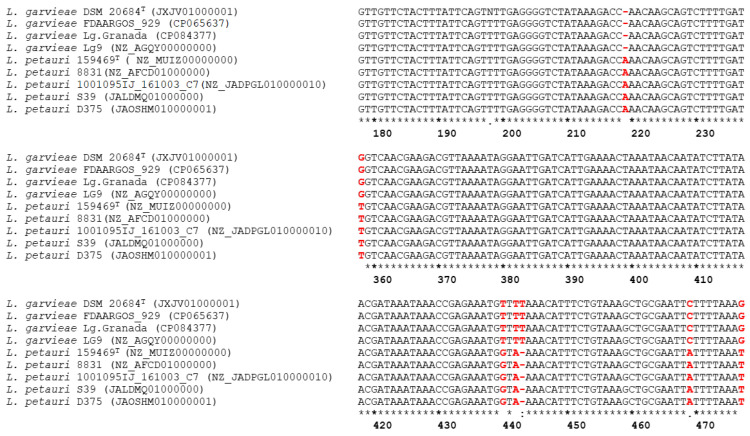
Multiple 16S-23S ITS sequences alignment of the *L. garvieae* and *L. petauri* control strains used in this study (accession number in parentheses). Polymorphism positions refer to the strain *L. garvieae* LMG9472 (accession number HM241914.1).

**Figure 2 microorganisms-11-01320-f002:**
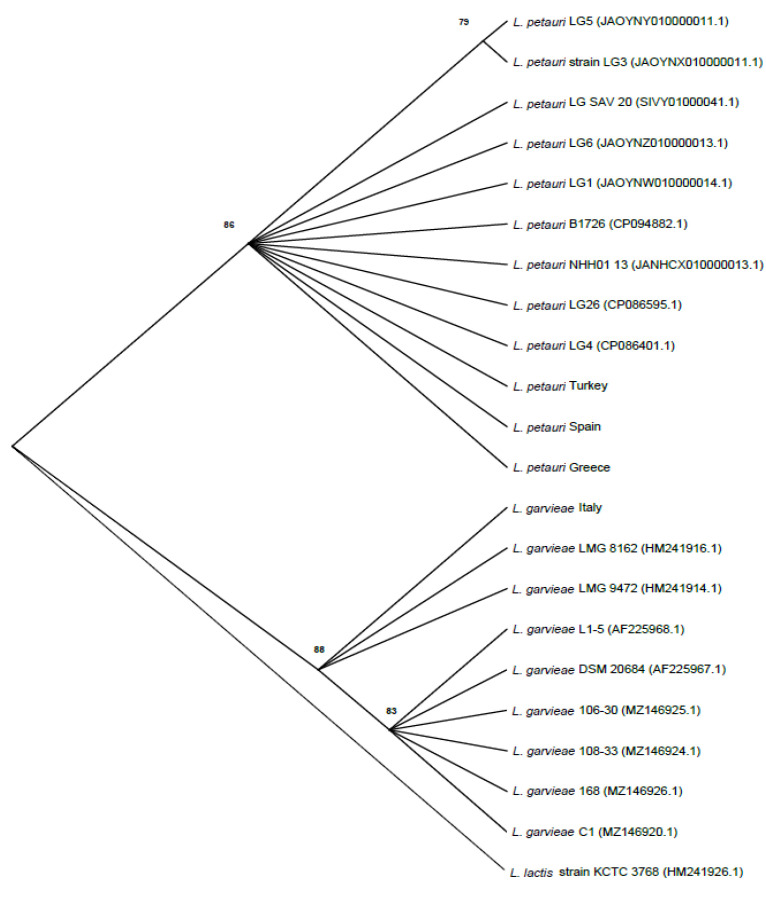
Evolutionary relationship of taxa based on an *ITS* analysis. The neighbor-joining method was used to infer the evolutionary history. The optimal tree is shown. The percentage of replicate trees in which the associated taxa clustered together in the bootstrap test (1000 replicates) is shown next to the branches. The evolutionary distances were computed using the Maximum Composite Likelihood method and are in the units of the number of base substitutions per site. This analysis involved 22 nucleotide sequences. All ambiguous positions were removed for each sequence pair (pairwise deletion option). There were a total of 535 positions in the final dataset. Evolutionary analyses were conducted in MEGA X. *Lactococcus lactis* strain KCTC 3768 was used as the outgroup. Strains reported in the tree as *L. petauri* Turkey, *L. petauri* Greece, and *L. petauri* Spain are each one representative of 20 different clinical strains per country. The *L. garvieae* Italy strain reported in the tree is representative of 22 different clinical strains.

**Table 1 microorganisms-11-01320-t001:** Field clinical strains isolated in lactococcosis outbreaks that occurred in rainbow trout farms in Italy, Spain, Greece, and Turkey.

Italy			Spain			Greece			Turkey		
Strain ID	Geographical Origin	Date of Isolation	Strain ID	Geographical Origin	Date of Isolation	Strain ID	Geographical Origin	Date of Isolation	Strain ID	Geographical Origin	Date of Isolation
1-IT	Quinto	2016	5239-VISAVET	Granada	2019	LG1	Macedonia	2016	Y-LG1	Trabzon	2016
2-IT	Quinto	2016	5424-VISAVET	Granada	2017	2-ELGO	Macedonia	2010	Y2-KTU	Gumushane	2016
3-IT	Cassolnovo	2016	5664-VISAVET	Granada	2017	LG3	Macedonia	2010	Y3-KTU	Trabzon	2016
4-IT	Cassolnovo	2016	5787-VISAVET	Granada	2016	4-ELGO	Macedonia	2010	Y6-KTU	Trabzon	2016
5-IT	Cerano	2017	02/6071-VISAVET	Granada	2002	LG5	Ipiros	2009	Y7-KTU	Trabzon	2017
6-IT	Cassolnovo	2017	8666-VISAVET	Granada	2016	LG6	Macedonia	2008	K2-KTU	Trabzon	2017
7-IT	Cassolnovo	2018	8059-VISAVET	Lérida	2016	7-ELGO	Ipiros	2009	K3-KTU	Gumushane	2016
8-IT	Cassolnovo	2018	8495-VISAVET	Madrid	2016	8-ELGO	Macedonia	2009	K7-KTU	Gumushane	2018
9-IT	Cassolnovo	2018	8516-VISAVET	La Coruña	2016	9-ELGO	Ipiros	2008	K8-KTU	Gumushane	2018
10-IT	Quinto	2019	03/8568-VISAVET	Asturias	2003	10-ELGO	Macedonia	2009	Kürtün-KTU	Gumushane	2020
11-IT	Quinto	2019	818-VISAVET	Guadalajara	2016	11-ELGO	Macedonia	2007	Rize-KTU	Rize	2020
12-IT	Cerano	2019	820-VISAVET	Guadalajara	2016	12-ELGO	Macedonia	2010	Antalya-KTU	Antalya	2019
13-IT	Cassolnovo	2019	393-VISAVET	Granada	2019	13-ELGO	Ipiros	2010	Vakfıkebir-KTU	Trabzon	2020
14-IT	Quinto	2020	195-VISAVET	La Coruña	2019	14-ELGO	Ipiros	2010	LG10	Mugla	2017
15-IT	Preore	2020	ICM16/00935	Granada	2016	15-ELGO	Ipiros	2008	20-KTU	Mugla	2016
16-IT	Ormelle	2017	307-VISAVET	Granada	2016	16-ELGO	Ipiros	2006	9-KTU	Kayseri	2018
17-IT	Cassolnovo	2017	1008-VISAVET	La Coruña	2017	17-ELGO	Ipiros	2006	13-KTU	Izmir	2016
18-IT	Quinto	2019	8831-VISAVET	Lérida	2017	18-ELGO	Macedonia	2007	123-KTU	Izmir	2017
19-IT	Cassolnovo	2019	P04/8864-VISAVET	Granada	2004	19-ELGO	Macedonia	2008	140-KTU	Elazig	2016
20-IT	Quinto	2019	8943-VISAVET	Granada	2016	20-ELGO	Ipiros	2007	107B-KTU	Rize	2019
1683	San Daniele Friuli	1997									
1691-2	Porcia	1997									

**Table 2 microorganisms-11-01320-t002:** Nucleotide differences in the 16S-23S rRNA internal transcribed spacer regions (*ITS*) of *L. garvieae* and *L. petauri*.

	16S-23S Region Position ^a^						
	219	358	440	442	443	469	478
**Strain type**							
*L. garvieae* DSM20684^T^	-	G	T	T	T	C	G
*L. petauri* 159469^T^	A	T	G	A	-	A	T
**Control strains**							
*L. garvieae* Lg.Granada	-	G	T	T	T	C	G
*L. garvieae* Lg 9	-	G	T	T	T	C	G
*L. garvieae* FDAARGOS_929	-	G	T	T	T	C	G
*L. petauri* 1001095IJ_161003_C7	A	T	G	A	-	A	T
*L. petauri* 8831	A	T	G	A	-	A	T
*L. petauri* D375	A	T	G	A	-	A	T
*L. petauri* S39	A	T	G	A	-	A	T
**Reference strains**							
*L. garvieae* LMG 9472	-	G	T	T	T	C	G
*L. garvieae* LMG 8162	-	G	T	T	T	C	G
*L. garvieae* C1	-	G	T	T	T	C	G
*L. garvieae* 108-33	-	G	T	T	T	C	G
*L. garvieae* 106-30	-	G	T	T	T	C	G
*L. garvieae* 168	-	G	T	T	T	C	G
*L. garvieae* 20684	-	G	T	T	T	C	G
*L. garvieae* L1-5	-	G	T	T	T	C	G
*L. petauri* LG4	A	T	G	A	-	A	T
*L. petauri* LG26	A	T	G	A	-	A	T
*L. petauri* B1726	A	T	G	A	-	A	T
*L. petauri* NHH01_13	A	T	G	A	-	A	T
*L. petauri* LG_SAV_20	A	T	G	A	-	A	T
**Clinical field strains**							
Italian (*n* = 22) ^b^	-	G	T	T	T	C	G
Spanish (*n* = 20) ^c^	A	T	G	A	-	A	T
Greek (*n* = 20) ^c^	A	T	G	A	-	A	T
Turkish (*n* = 20) ^c^	A	T	G	A	-	A	T

^a^ Positions of reported polymorphisms were inferred from the 16S-23S ribosomal RNA intergenic spacer sequence of the strain *L. garvieae* LMG 9472 (accession number HM241914.1). ^b^ This SNP pattern was consistently detected in all the Italian field clinical strains. ^c^ This SNP pattern was consistently detected in all the Spanish, Greek, and Turkish field clinical strains.

**Table 3 microorganisms-11-01320-t003:** ANI and dDDH values (%) of the control and the subset of clinical strains examined in this study.

Isolate	*L. petauri* 159469^T^ (NZ_MUIZ0000000)		*L. garvieae* DSM 20684^T^ (JXJV01000001)		Accession Number
	%ANI	%DDH	%ANI	%DDH	
**Control strains**					
1001095IJ_161003_C7	**98.8**	**89.5**	93.1	50.9	NZ_JADPGL010000010
8831	**98.5**	**85.5**	93.0	50.5	NZ_AFCD01000000
D375	**98.4**	**85.4**	93.1	50.7	JAOSHM010000001
S39	**98.3**	**85.5**	93.0	50.8	JALDMQ010000000
Lg. Granada	93.2	51.7	**98.3**	**85.1**	CP084377
Lg 9	93.2	51.2	**98.3**	**85.3**	NZ_AGQY00000000
FDAARGOS_929	92.9	50.8	**99.9**	**99.9**	CP065637
**Field clinical strains**					
ICM16/00935 ^a^	**98.4**	**85.7**	92.0	50.5	JARHWC000000000
1683 ^b^	93.3	51.1	**98.4**	**85.4**	JARHWV000000000
1691-2 ^b^	93.3	51.1	**98.4**	**85.4**	JARHWU000000000
Y-LG1 ^c^	**98.4**	**85.6**	93.0	50.8	JAQPON000000000
LG10 ^c^	**98.4**	**85.5**	93.1	50.8	JAQPOM000000000
LG6 ^d^	**98.5**	**85.4**	93.1	50.4	JAOYNZ010000001
LG5 ^d^	**98.4**	**85.4**	92.4	50.4	JAOYNY010000001
LG3 ^d^	**98.4**	**85.4**	92.5	50.4	JAOYNX010000001
LG1 ^d^	**98.4**	**85.3**	92.9	50.5	JAOYNW010000001

^a, b, c, d^ Spanish, Italian, Turkish, and Greek clinical field strains, respectively.

## Data Availability

A single representative ITS sequence for a strain, isolated in different countries, was deposited in GenBank (www.ncbi.nlm.nih.gov/genbank (accessed on 2 January 2023)) under the following accession numbers: 1-IT for Italian strains OQ108343; 1-GR for Greek strains OQ108344; 1-TK for Turkish strains OQ108345; and 1-SP for Spanish strains OQ108346.

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
