# Peer review of "16S-23S rRNA Internal Transcribed Spacer Region (ITS) Sequencing: A Potential Molecular Diagnostic Tool for Differentiating Lactococcus garvieae and Lactococcus petauri"

_microorganisms, 2023, doi:10.3390/microorganisms11051320_

Round 1

Reviewer 1 Report

The authors proposed a new fast test to distinguish between 2 pathogenic for trouts Lactococcus sp.;L.garvieae and L.petauri 

Biochemical  profiles and genomes of both species showed a high degree of similarity and thus , classic diagnostic tests cannot distinguish these two species. The authors tried to develop a rapid, inexpensive  diagnostic test .

In this vein the  transcriped Spacer (ITS) region between 32 16S rRNA and 23S rRNA regions of 80 Lactococcus strains was amplified and sequenced. The 16S-23S rRNA ITS region permit to distinguish  between these 2 species and was proposed to be used as diagnostic marker to quickly identify the pathogens in a lactococcosis outbreak.

it is a well written and scientific relevant paper based on a solid bibliography and well design methodology that could be used as a diagnostic tool in case of outbreaks 

my suggestion is to ACCEPT and publish the paper 

Author Response

We appreciate the time and effort the reviewer dedicated to our manuscript and the comments provided.

Reviewer 2 Report

Stoppani et al proposed a work about the development of a faster and cheaper method for distinguish Lactococcus garvieae from Lactococcus petauri, since currently available tools are genomic methods which require high amount of work and are expensive.

The main problem of this work is how the the seven SNPs, that should distinguish between the two strains, are identified. Stoppani et al started from 80 strain that were 79 garvieae and only 1 petauri. Then they amplified a single region and found that 5 SNPs and 2 indels are different between samples downloaded from Genbank. Based on such 7 differences, they surprisingly conclude that the identification of their 80 samples is wrong and that such 7 difference can distinguish between garvieae and petauri. There isn't any kind of support on this conclusion, since they have checked only 1000 bases in a genome (about 2 Mb) that is 2 thousand times longer that such small regione. Basically, authors did some kind of GWAS: they have 2 different "phenotypes" (=being "garvieae" or "petauri"), they found out SNPs/i and they basically claimed that the SNPs are useful and enough to distinguish between the two strains despite the fact that strains are classified differently! They is no proof that those seven SNPs/indels found can be used to discriminate between the two strains. This is a methodological error that basically affect negatively the entire work and made it unpublishable.

Reviewer 3 Report

The paper described sequence analysis of the 16S-23S rRNA ITS region from 80 Lactococcus strains. The results suggested that the ITS region has enough resolution to differentiate the two closely related Lactococcus garvieae from Lactococcus petaurid, and that the ITS can be used as diagnostic marker to quickly identify the pathogens in a lactococcosis outbreak.

Major comment: The title and the study aim are both misleading; sequencing analysis is not a fast diagnostic technique as it is time consuming and involves a multitude of steps like other molecular methods (i.e., DNA extraction, PCR, purification of amplicons, cycle sequencing, DNA sequencing, then analysis). It requires costly instrumentation and reagents, and so not cheap. The aim also stated development of a diagnostic method, but the use of ITS sequence for differentiating closely related species has been around for ages. The reviewer recommends modifying the title and the aim, into suggesting that the ITS region is a potential molecular target for differentiating those two species of Lactococcus.

Author Response

Reviewer 3 Comments and Suggestions for Authors

 The paper described sequence analysis of the 16S-23S rRNA ITS region from 80 Lactococcus strains. The results suggested that the ITS region has enough resolution to differentiate the two closely related Lactococcus garvieae from Lactococcus petaurid, and that the ITS can be used as diagnostic marker to quickly identify the pathogens in a lactococcosis outbreak.

 Major comment: The title and the study aim are both misleading; sequencing analysis is not a fast diagnostic technique as it is time consuming and involves a multitude of steps like other molecular methods (i.e., DNA extraction, PCR, purification of amplicons, cycle sequencing, DNA sequencing, and further analysis). It requires costly instrumentation and reagents, and so not cheap. The aim also stated development of a diagnostic method, but the use of ITS sequence for differentiating closely related species has been around for ages. The reviewer recommends modifying the title and the aim, into suggesting that the ITS region is a potential molecular target for differentiating those two species of Lactococcus.

We appreciate the time and effort the reviewer dedicated to our manuscript providing insightful feedback on ways to strengthen our paper.

The adjectives fast and cheap were not used with an absolute value but referred to the use of genomes comparisons of the two species for diagnostic purposes. We agree too that the use of ITS region for diagnostic purpose it is not novel, and ITS sequence analysis for proximate species differentiation has been used for many years. We removed the sentence from the conclusion as reported below.

Moreover, following the referee suggestions changing the title and the aim of the manuscript.

Title

16S-23S rRNA Internal Transcribed Spacer region (ITS) sequencing: a potential molecular diagnostic tool for differentiating Lactococcus garvieae and Lactococcus petauri.

Abstract: Lactococcus garvieae is the etiological agent of lactococcosis, a clinically and economically significant infectious disease affecting farmed rainbow trout. L. garvieae has been considered the only cause of lactococcosis for a long time; however, L. petauri, another species of the genus Lactococcus, has lately been linked to the same disease. The genomes and biochemical profiles of L. petauri and L. garvieae have a high degree of similarity. Traditional diagnostic tests currently available do not distinguish these two species. The aim of this study was to apply Transcribed Spacer (ITS) region between 16S rRNA and 23S rRNA as potential useful molecular target to differentiate L. garvieae from L. petauri, saving time and money compared to genomics methods, actually reported as diagnostic tool for accurate discrimination between these two species. ITS region of 80 strains was amplified and sequenced. The amplified fragments varied in size from 500 to 550 bp. Based on the sequence seven SNPs were identified that separate L. garvieae from L. petauri. The 16S-23S rRNA ITS region has enough resolution to distinguish between closely related L. garvieae and L. petauri, and it can be used as diagnostic markers to quickly identify the pathogens in a lactococcosis outbreak.

  1. Discussion
  2. garvieae had been considered until recently the only etiological agent to lactococcosis [1]. However, recent studies have demonstrated the implication of the new lactococcal species, L. petauri, in the etiology of this disease [4-7]. Therefore, both L. petauri and L. garvieae are responsible for lactococcosis outbreaks. Differentiation of both species in routine of microbiological diagnostic laboratories is difficult due to their phenotypic and genetic similarity, which resulted in identification errors. Thus, commercial identification systems such as MALDI-TOF, or PCR assays targeting the 16S rRNA gene cannot discriminate between isolates of both species [7, 13,14].Nevertheless, the interspecies divergence in the 16S–23S ITS sequences makes this region a potentially good candidate for bacterial discrimination and identification. Therefore, in this study we investigate the polymorphism in the 16S rRNA and 23S rRNA Transcribed Spacer (ITS) region as potential useful molecular target to differentiate L. garvieae from L. petauri.

As there was the possibility that the identification of some of the L. garvieae and L. petauri strains available in public databases was not correct, the identification of a subset of 7 strains (control strains) was corroborated by calculating their dDDH and ANI values by comparing their genomes with those of the L. garvieae and L. petauri type strains. Control strains displayed dDDH and ANI values higher than the threshold values considered for species delineation, confirming therefore their correct identification (Table 2). Their 16S-23S ITS sequences were aligned revealing two different ITS patterns for L. garvieae and L. petauri and characterized by differences in 5 SNP and 2 indel variations (Table 2). Later on, these variations were confirmed in all the reference and field clinical strains when their sequences were matched to those of the control strains (Table 2). Thus, reference strains exhibited an ITS pattern according to their identification. Regarding clinical field strains, Italian isolates exhibited the ITS pattern characteristic of L. garvieae, while the Spanish, Greek and Turkish isolates displayed the ITS pattern of L. petauri. These results are consistent with previous results based on WGS and confirm the relevant role of L. petauri in the epidemiology of lactococcosis in some Mediterranean countries [5, 13]. Overall, the results of this study indicate the great potential of the identified SNP markers for the differentiation of L. garvieae from L. patauri. Phylogenetic analysis based on the sequencing of the gyrB gene, already used as a phylogenetic marker for several closely related genera [28], can discriminated between L. petauri and L. garvieae, grouping isolates of these species in different clusters [7]. Similarly, phylogenetic analysis based on the 16S-23S sequences grouped the L. garvieae and L. petauri strains in two well defined clusters (Fig. 2). In this sense, sequencing of the 16S-23S ITS regions should be as accurate, efficient and cost-effective as the gyrB sequencing that it has also been proposed for differentiating both species [6, 7].

  1. Conclusions

The results of this study show that L. garvieae and L. petauri strains display two well differentiated patterns in their the 16S-23S ITS region, characterized by five SNP differences and two indel variations that were consistently identified in all the strains investigated. Therefore, sequencing of the 16S-23S rRNA ITS region can be a useful tool for discriminating between L. garvieae and L. petauri.

Round 2

Reviewer 2 Report

Although now the metodology improved by finding those discriminating 5 SNPs and 2 indels using a dataset with L. petauri and L. garvieae sequences well classified (and not using 80 sequences for which the results is used to say that they are wrongly classified), a main problem on metodology remains. The 7 differences found could be discriminating, but without a proper GWAS analysis is not possibile to know wheter theay are truly discriminating and how much is their power. So please add a GWAS analysis to proper calculate probability about the pattern is not random, which will take into account the number of sequences analyzed (which could not be so high) and the regione length.
